# Hierarchical Vector Quantized Graph Autoencoder with Annealing-Based Code Selection

## ABSTRACT

Graph self-supervised learning has gained significant attention recently. However, many existing approaches heavily depend on perturbations, and inappropriate perturbations may corrupt the graph's inherent information. The Vector Quantized Variational Autoencoder (VQ-VAE) is a powerful autoencoder extensively used in fields such as computer vision; however, its application to graph data remains underexplored. In this paper, we provide an empirical analysis of vector quantization in the context of graph autoencoders, demonstrating its significant enhancement of the model's capacity to capture graph topology. Furthermore, we identify two key challenges associated with vector quantization when applying in graph data: codebook underutilization and codebook space sparsity. For the first challenge, we propose an annealing-based encoding strategy that promotes broad code utilization in the early stages of training, gradually shifting focus toward the most effective codes as training progresses. For the second challenge, we introduce a hierarchical two-layer codebook that captures relationships between embeddings through clustering. The second layer codebook links similar codes, encouraging the model to learn closer embeddings for nodes with similar features and structural topology in the graph. Our proposed model outperforms 16 representative baseline methods in self-supervised link prediction and node classification tasks across multiple datasets. Our implementation is available at https://anonymous.4open.science/r/hqa-gae-D2F4.

## KEYWORDS

Do, Not, Us, This, Code, Put, the, Correct, Terms, for, Your, Paper

**ACM Reference Format:**
Anonymous Author(s). 2018. Hierarchical Vector Quantized Graph Autoencoder with Annealing-Based Code Selection. In *Proceedings of Make sure to enter the correct conference title from your rights confirmation emai (Conference acronym 'XX).* ACM, New York, NY, USA, 11 pages. https://doi.org/XXXXXXX.XXXXXXX

## 1 INTRODUCTION

Graphs are prevalent in the real world, which are widely used to model the complex relationships between entities in systems like social networks and the web. However, the non-Euclidean nature and the scarcity of labeled data pose significant challenges in graph analysis. Recently, graph self-supervised learning (SSL) has been

proposed to address these issues by uncovering meaningful patterns from massive unlabeled data through pretext tasks. Graph SSL has been demonstrated to be useful in a wide range of downstream applications, such as social recommendation [42] and molecular property prediction [17].

Recently, graph contrastive learning (GCL) has been a dominant approach for self-supervised learning on graphs. Existing studies on GCL mainly rely on perturbing the original graph information to generate new views. However, it has been pointed out in [51] that inappropriate perturbations could disrupt the graph's inherent structure, leading to information loss or noise corruption. In other words, poorly-designed view generation strategies could degrade model performance, resulting in learned embeddings that lack semantic consistency. Therefore, the design of suitable perturbation techniques is crucial for these models, adding difficulty to their development.

Another prominent approach for implementing SSL on graphs is graph autoencoding. One popular and effective variant is masked graph autoencoding, which utilizes masked node features [15] or graph topology [22] to learn robust node representations. However, similar to contrastive learning, these masking methods can also introduce inappropriate perturbations, risking the loss of the graph's inherent information. Beyond perturbation-based methods, Vector Quantized Variational Autoencoders (VQ-VAE) [38] have emerged as an alternative approach, which encodes input features into discrete latent embeddings by mapping them into a quantized codebook and decodes by retrieving corresponding codebook entries to reconstruct the raw input features. It offers significant data compression capabilities that enable the model to be easily applied to large-scale data. Despite the success, its application to graph data remains underexplored. Existing works either focus on limited downstream tasks like molecular graph classification [6, 43], or rely on supervised training [45], falling far short in fully leveraging VQ-VAE for graph SSL.

In this paper, to bridge the gap, we extend VQ-VAE to graph SSL. We first empirically analyze VQ-VAE when applied to graph data, showing how well *vector quantization* can enhance the model's capability in capturing the underlying graph topology. Then we identify two key challenges when applying VQ-VAE to graph:

The first issue is *codebook space underutilization* arising from the "winner-take-all" principle in competitive learning [1, 9], where many discrete codes within the codebook remain unused during the training process. This insufficiency limits the model's capacity to represent diverse feature patterns in the graph. While Gumbel-Softmax [18] has been explored as a potential solution to improve codebook utilization by enabling more flexible sampling, our experiments on graph data indicate that it produces less satisfactory results, possibly due to the randomness introduced by reparameterization, which increases the instability of gradient updates. To address this limitation, we propose an annealing-based encoding

strategy, which dynamically selects code embeddings in the codebook. Specifically, in the early stage, the model is encouraged to explore a wide range of available code vectors, which forces the model to utilize more codes. With training epochs, the probability of selecting useless codes decreases and the model will concentrate more on the effective ones.

The second challenge is *codebook space sparsity*, which refers to the fact that in traditional VQ-VAE, individual code vectors are treated as independent entities with no regard for the inherent relationships between nodes in the graph. This may lead to the projection of similar nodes into different code vectors. To overcome the issue, we introduce a second layer on top of the first one, developing a two-layer codebook with a hierarchical structure that reflects the relationships between codes. The second layer can connect codes with similar embeddings, which can be used to further promote close embedding learned for similar nodes in the graph. Finally, our main contributions in this paper are summarized as follows.

- We propose a novel graph SSL method HQA-GAE, which is a **H**ierarchical vector **Q**uantized and **A**nnealing code selection based **G**raph **A**uto-**e**ncoder.
- We qualitatively analyze the effectiveness of vector quantization in utilizing graph topology and experimentally verify our claim.
- We present two key challenges in applying VQ-VAE to graph SSL: codebook space underutilization and codebook space sparsity. We further put forward an annealing-based code selection strategy and a hierarchical codebook mechanism to solve the issues, respectively.
- We conduct extensive experiments to demonstrate the superior performance of HQA-GAE over 16 other state-of-the-art methods on both node classification and link prediction tasks.

## 2 RELATED WORK

This section reviews recent advances on graph SSL, with a focus on two primary approaches: graph contrastive learning and autoencoding techniques.

*Graph Contrastive Learning (GCL)* has emerged as a promising approach for SSL on graph-structured data. It aims to learn robust node or graph-level representations by maximizing the agreement between different augmented views of the same node/graph, while minimizing that with views of other nodes/graphs. Although early works [25, 35, 41, 46, 47, 50] have demonstrated their efficacy, a key limitation of GCL lies in the reliance on manually designed augmentations that are often task-specific and may disrupt the structural integrity of graphs, leading to suboptimal performance in certain domains [47]. Further, poorly designed augmentation schemes can inadvertently introduce noise, diminishing the semantic consistency of the learned embeddings [51].

*Graph auto-encoding* is another technique for graph SSL, which learns node embeddings by reconstructing the given input graph. In addition to graph auto-encoders [20, 27] and variational graph auto-encoders [12, 20, 23, 27], some advanced models have recently been proposed. For example, *Masked Graph Autoencoders (MGAE)*, such as GraphMAE [15] and MaskGAE [22], have drawn significant attention. These methods [14, 36, 49] introduce masking strategies for graph features or edges, followed by reconstruction, and have shown promising results. Despite the success, the effectiveness of

masked autoencoding heavily relies on the choice of masking strategies. Inappropriate masking can lead to significant information loss, which degrades the model performance.

Further, *Vector Quantized Variational Autoencoders (VQ-VAE)* have demonstrated notable success in the fields of computer vision [7, 30, 38] and computer audition [5, 48] by discretizing latent spaces, enhancing robustness and efficiency. They also hold potential for self-supervised learning on graphs; however, existing applications remain limited and often fail to fully leverage the VQ-VAE framework. The early attempt VQ-GNN [6] explores the use of vector quantization as a dimensionality reduction tool in GNNs, which deviates significantly from the original VQ-VAE training scheme. Mole-BERT [43] and DGAE [4] apply VQ-VAE for molecular graph classification but restrict its scope to specific domains, lacking generalizability to broader graph tasks like node classification and link prediction. VQ-Graph [45], on the other hand, adopts the VQ-VAE training approach but introduces labeled data rather than pursuing SSL. Moreover, key challenges, such as codebook underutilization and space sparsity, have not been adequately addressed in previous works, which hinder model performance. Instead, our work aims to directly tackle these issues by exploring the capabilities of VQ-VAE in graph data, thereby enhancing the effectiveness of graph representation learning.

## 3 PRELIMINARIES

### 3.1 Graph Self-supervised Learning

A graph $\mathcal{G} = (\mathcal{V}, \mathcal{E})$ consists of a set of nodes $\mathcal{V}$ and edges $\mathcal{E}$, where $|\mathcal{V}| = N$ and each node $v_i \in \mathcal{V}$ can be associated with a feature vector $\mathbf{x}_i \in \mathbb{R}^D$. The adjacency matrix $\mathbf{A} \in \mathbb{R}^{N \times N}$ encodes the connectivity of the graph, where $\mathbf{A}_{ij} = 1$ if an edge exists between nodes $v_i$ and $v_j$, and 0 otherwise. Self-Supervised Learning (SSL) on graphs [24, 44] aims to learn useful representations without requiring labeled data. By leveraging pretext tasks, such as node feature reconstruction or contrastive learning, the model is encouraged to learn semantic embeddings $\mathbf{h}_i \in \mathbb{R}^d$ for each node, which capture key graph properties.

### 3.2 Graph Neural Network

Graph Neural Networks (GNNs) are powerful tools for learning from graph-structured data. A widely used GNN framework is the Message Passing Neural Network (MPNN) [3, 19], which iteratively updates node representations based on messages passed from neighboring nodes. In the $k$-th message-passing iteration, a node $v_i$'s representation $\mathbf{h}_i^{(k)}$ is updated by aggregating the features of its neighbors $\mathcal{N}(i)$, starting from an initial representation $\mathbf{h}_i^{(0)} = \mathbf{x}_i$. This process can be described by:

$$\mathbf{h}_i^{(k)} = \sigma\left(\mathbf{h}_i^{(k-1)}, \bigoplus_{j \in \mathcal{N}(i)} \phi^{(k)}\left(\mathbf{h}_i^{(k-1)}, \mathbf{h}_j^{(k-1)}\right)\right), \quad (1)$$

where $\bigoplus$ is a permutation-invariant aggregation function (e.g., mean, sum, or max), $\phi$ denote differentiable functions like linear projections, and $\sigma$ is a non-linear function. The output after $K$ layers of message passing, $\mathbf{h}_i^{(K)}$, serves as the final representation for node $v_i$.

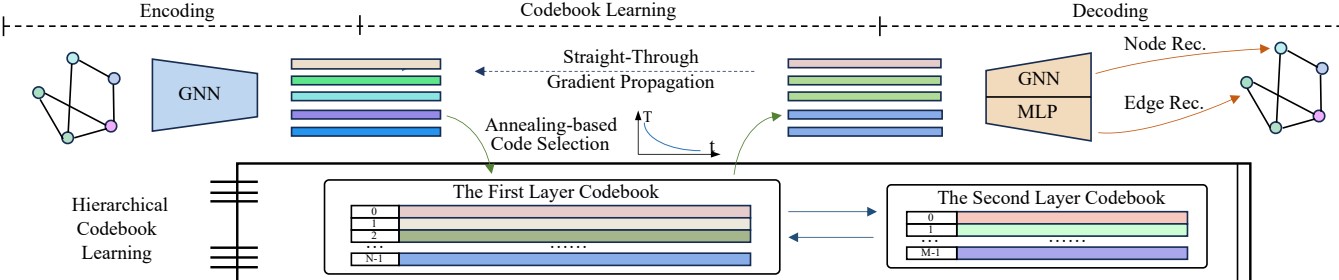

**Figure 1: The schematic diagram of HQA-GAE, including graph encoding and decoding, annealing-based code selection, hierarchical codebook learning.**

### 3.3 VQ-VAE

VQ-VAE offers a perturbation-free autoencoder for learning discrete representations, which encodes input features into discrete latent embeddings by mapping them into a quantized codebook and decodes by retrieving corresponding codebook entries to reconstruct the raw input features. Specifically, the encoder $E$ maps input $\mathbf{x}_i$ to a latent vector $E(\mathbf{x}_i)$, which is then quantized to the most similar code embedding $\mathbf{e}_i$, where $i = \arg\max_j \text{sim}\left(E(\mathbf{x}_i), \mathbf{e}_j\right)$, and the function $\text{sim}\left(E(\mathbf{x}_i), \mathbf{e}_j\right)$ denotes the similarity between the latent vector $E(\mathbf{x}_i)$ and the codebook embedding $\mathbf{e}_j$. This similarity can be quantified using various metrics, such as cosine similarity or negative Euclidean distance.

The total loss of VQ-VAE comprises three terms: (1) reconstruction loss, ensuring that the decoder $D$ accurately reconstructs the original input $\mathbf{x}_i$; (2) commitment loss, which forces the encoder to stabilize by minimizing the gap between $E(\mathbf{x}_i)$ and the selected codebook embedding $\mathbf{e}_i$; and (3) codebook loss, updating the codebook entries to align with the encoder's output. Formally, we have:

$$\mathcal{L} = \frac{1}{N} \sum_i^N \Big( \|\mathbf{x}_i - D(\mathbf{e}_i)\|_2^2 \tag{2}$$
$$+ \|\text{sg}(E(\mathbf{x}_i)) - \mathbf{e}_i\|_2^2 + \eta \|\mathbf{e}_i - \text{sg}(E(\mathbf{x}_i))\|_2^2 \Big),$$

where $\text{sg}(\cdot)$ is the stop-gradient operator, $\eta$ is a balance weight, and $N$ is the number of training samples. In VQ-VAE, during the gradient descent optimization of the reconstruction loss, the encoder's gradients are directly copied from the decoder. From a clustering view, VQ-VAE inherently facilitates grouping similar data points by quantizing them to the same codebook entries. Each codebook entry acts as a cluster center for hidden embeddings of data points.

## 4 METHOD

In this section, we first introduce VQ-GAE as the starting point of our approach, highlighting the role of vector quantization in enhancing the model's capacity to capture graph topological structure. Building on VQ-GAE, we next incorporate **annealing-based code selection** and **hierarchical codebook design**, which together form the core of our HQA-GAE. Finally, we elaborate on the encoder and decoder models, as well as the training objectives. The overall framework of HQA-GAE is illustrated in Figure 1.

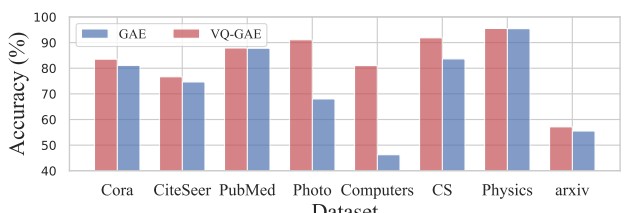

**Figure 2: The performance of node classification tasks for both normal GAE and VQ-GAE, where an MLP is used as the encoder and the decoder remains unchanged as a GNN.**

### 4.1 Starting with VQ-GAE

The vanilla Vector Quantized Graph Autoencoder (VQ-GAE) first encodes node $v_i$'s feature vector $\mathbf{x}_i$ into a continuous representation $\mathbf{h}_i$ via an encoder. Let the embedding space be further represented by a set of code embeddings $\{\mathbf{e}_j\}_{j=1}^M$ in a codebook, where $M \ll N$ and $N$ denotes the number of nodes in the graph. The continuous representation $\mathbf{h}_i$ is then quantized by selecting the most appropriate code embedding by:

$$\mathbf{e}_i = \text{lookup}\left(\mathbf{h}_i, \{\mathbf{e}_j\}_{j=1}^M\right), \tag{3}$$

where $\text{lookup}(\cdot)$ denotes the function that retrieves the code embedding with the highest similarity to $\mathbf{h}_i$. A detailed discussion on the lookup function will be provided in Section 4.2. The quantized representation $\mathbf{e}_i$ is subsequently utilized to reconstruct the raw node feature $\mathbf{x}_i$, while the hidden representation $\mathbf{h}_i$ is used to reconstruct the adjacency matrix $\mathbf{A}$. In typical VQ-VAE methods applied to image data, an image is represented by multiple code embeddings, where the space is geometrically divided into patches, each of which is represented by a separate code embedding. However, in VQ-GAE, each node is represented by a single code embedding, due to the lack of a spatial structure in a single node feature. **Note that since the codebook size is much smaller than the number of nodes, we can only use the output of encoder $\mathbf{h}_i$ instead of the code embedding $\mathbf{e}_i$ as the final node representation.**

While the code embedding $\mathbf{e}_i$ is not the final representation, vector quantization still plays a crucial role in enhancing the model's capability to capture the graph's topology. Specifically, for nodes with similar features that are encoded into the same codebook

embedding, vector quantization forces the model to leverage the structural difference when reconstructing their raw features. This in turn positively affects the encoder to inject more useful topological information into the learned representations for nodes. In the case of VQ-GAE, the optimization process employs the straight-through gradient propagation mechanism[1], which allows the gradient to be directly passed from the decoder to the encoder. This process encourages the encoder to integrate more relevant topological information, which is conveyed by the decoder, into the learned representations.

To verify the claim, we conduct a preliminary series of experiments by comparing the performance of the VQ-GAE with that of a normal GAE, which removes the vector quantization module between the encoder and decoder. In both models, we use a simple Multi-Layer Perceptron (MLP) as encoder that only takes node features as input to prevent explicitly learning graph structural information in node embeddings. As depicted in Figure 2, VQ-GAE performs better than GAE across various benchmark datasets, which shows that incorporating a codebook into GAE can significantly improve the model performance, even when the encoder does not explicitly capture topological information.

## 4.2 Annealing-based Code Selection

Typically, VQ-GAE uses an argmax-based lookup function when selecting the code with the highest similarity to the input node embedding. The selected embedding $\mathbf{e}_i$ from the codebook becomes increasingly closer to input $\mathbf{h}_i$ over iterations, which increases its probability to be selected in the future. This phenomenon results in a "winner-take-all" effect, where selection bias amplifies, exacerbating the codebook space underutilization problem. The underutilization leads to over-reliance on a few frequently used code embeddings, which hinders generalization to diverse data distributions and results in the loss of certain detailed information.

To mitigate the problem, we replace the argmax-based lookup function with a softmax-based probabilistic function. Specifically, for each node $v_i$, we first compute the similarity $s_{i,j} = \text{sim}(\mathbf{h}_i, \mathbf{e}_j)$ between $\mathbf{h}_i$ and code embedding $\mathbf{e}_j$, and then use softmax to calculate the probability $p_{i,j}$ of selecting $\mathbf{e}_j$:

$$p_{i,j} = \frac{\exp(s_{i,j}/T)}{\sum_i \exp(s_{i,j}/T)}, \qquad (4)$$

where $T$ is the temperature parameter that controls the smoothness of the distribution. A higher value of $T$ leads to a smoother, more uniform distribution, while a lower $T$ sharpens the selection. Fixing the temperature $T$ throughout this process, like Gumbel-Softmax, may restrict the model's capability to explore diverse solutions during training [21], potentially leading to trapping into local patterns too early. To address the problem, we adopt an annealing-based update strategy, where $T$ starts from an initial temperature $T_0$ and decays exponentially over iterations:

$$T_k = \max(\gamma T_{k-1}, \epsilon). \qquad (5)$$

Here $\gamma \in (0, 1)$ is the decay factor, and $\epsilon$ is a small positive constant that prevents numerical instability as $T$ approaches zero. During

early training, a higher $T$ promotes exploration of a wide range of code embeddings, encouraging the model to utilize more codes and avoid the winner-take-all problem. As training progresses and $T$ decreases, the probability of selecting less useful codes reduces, allowing the model to focus more on the effective ones.

## 4.3 Hierarchical Codebook Design

In VQ-GAE, individual codebook entries are treated as independent entities, disregarding the inherent relationships between codebook embeddings. This often leads to codebook space sparsity, where similar features are not sufficiently close in the representation space. To address the issue, we aim to interconnect codes in the codebook rather than treat them independently, aligning better with the characteristics of graph data. Specially, we introduce a second-layer codebook, which we refer to as *a codebook of the codebook*, to establish relationships among the codes.

Let the embedding of the $i$-th code in the first-layer codebook be denoted by $\mathbf{e}_{1,i} \in \{\mathbf{e}_{1,k}\}_{k=1}^{M}$, where $M$ is the total number of codes in the first-layer codebook. The embedding of the $j$-th code in the second-layer codebook is represented by $\mathbf{e}_{2,j} \in \{\mathbf{e}_{2,k}\}_{k=1}^{C}$, where $C < M$ and represents the number of codes in the second-layer codebook. The optimization objective, similar to k-means clustering [2], is then defined as:

$$O = \max \sum_{j=1}^{C} \sum_{i \in S_j} \text{sim}(\mathbf{e}_{1,i}, \mathbf{e}_{2,j}), \qquad (6)$$

where $S_j$ represents the set of codes $\mathbf{e}_{1,i}$ that are assigned to the second-layer code $\mathbf{e}_{2,j}$. The function $\text{sim}(\mathbf{e}_{1,i}, \mathbf{e}_{2,j})$ measures the similarity between the first-layer code $\mathbf{e}_{1,i}$ and the second-layer code $\mathbf{e}_{2,j}$. The optimization strategy for this training objective is to simultaneously update both codebook embeddings, encouraging similar embeddings in the first-layer codebook to cluster closely, while ensuring that dissimilar embeddings are split farther apart, thus reducing codebook space sparsity. The detailed optimization procedure will be discussed in Section 4.5.

## 4.4 Encoder and Decoder Models

For the encoder, our method allows for the use of various architectures, such as GCN [19], GraphSAGE [10], and GAT [40]. Further, the decoder is divided into two components: one for reconstructing node features and the other for reconstructing edges. For node feature reconstruction, we employ GAT as in [15]. For edge reconstruction, we adopt a methodology similar to MaskGAE [22], where a MLP and the dot product between node embeddings are used. Specifically, we define the structural decoder $D_{\text{edge}}$ as follows:

$$D_{\text{edge}}(\mathbf{h}_i, \mathbf{h}_j) = \text{Sigmoid}(\text{MLP}(\mathbf{h}_i \circ \mathbf{h}_j)), \qquad (7)$$

where $\mathbf{h}_i$ and $\mathbf{h}_j$ are node representations for $v_i$ and $v_j$ respectively from the encoder, and $\circ$ denotes the vector dot product.

## 4.5 Training Objective

Our loss function consists of both reconstruction loss and vector quantization loss (VQ loss). The reconstruction loss has two components: node reconstruction loss and edge reconstruction loss. For node reconstruction, we employ the scaled cosine error [15] to capture the difference in node features. Unlike VQGraph [45], which

---

[1]As mentioned in Section 3.3, since the process of selecting the appropriate embedding from the codebook is non-differentiable, during optimization, the gradient of the encoder in VQ-VAE is directly copied from the decoder.

reconstructs the entire adjacency matrix, we use negative sampling to compute the edge reconstruction loss due to the sparsity of edges. Specifically, the loss can be formulated as:

$$\mathcal{L}_{\text{NodeRec}} = \frac{1}{N} \sum_{i=1}^{N} \left( 1 - \frac{\mathbf{x}_i^T \hat{\mathbf{x}}_i}{\|\mathbf{x}_i\| \|\hat{\mathbf{x}}_i\|} \right)^{\lambda}, \text{ where } \hat{\mathbf{x}}_i = D_{\text{node}}(\mathbf{e}_{1,i}),$$

$$\mathcal{L}_{\text{EdgeRec}} = - \frac{1}{|\mathcal{E}^+|} \sum_{(v_i,v_j)\in\mathcal{E}^+} \log D_{\text{edge}}\left(\mathbf{h}_i, \mathbf{h}_j\right)$$
$$- \frac{1}{|\mathcal{E}^-|} \sum_{(v_{i'},v_{j'})\in\mathcal{E}^-} \log\left(1 - D_{\text{edge}}\left(\mathbf{h}_{i'}, \mathbf{h}_{j'}\right)\right),$$

$$(8)$$

where $N$ is the total number of nodes in the graph, and $\hat{\mathbf{x}}_i$ represents the reconstructed feature vector, obtained by decoding the code embedding $\mathbf{e}_{1,i}$ from the first-layer codebook through the node decoder $D_{\text{node}}$. The scaling factor $\lambda$ controls the sensitivity of the node reconstruction loss to feature discrepancies. $\mathcal{E}^+$ refers to the set of observed (positive) edges in the graph, while $\mathcal{E}^-$ represents a set of negative edges generated through negative sampling [22, 26]. The structural decoder is parameterized denoted as $D_{\text{edge}}(\cdot)$.

In addition to the reconstruction loss, the VQ loss comprises two components corresponding to the two layers of codebooks used in the vector quantization process:

$$\mathcal{L}_{\text{vq1}} = \frac{1}{N} \sum_{i=1}^{N} \left( \left\| \text{sg}\left[\mathbf{e}_{1,i}\right] - \mathbf{h}_i \right\|_2^2 + \left\| \text{sg}\left[\mathbf{h}_i\right] - \mathbf{e}_{1,i} \right\|_2^2 \right),$$

$$\mathcal{L}_{\text{vq2}} = \frac{1}{N} \sum_{i=1}^{N} \left( \left\| \text{sg}\left[\mathbf{e}_{2,i}\right] - \mathbf{e}_{1,i} \right\|_2^2 + \left\| \text{sg}\left[\mathbf{e}_{1,i}\right] - \mathbf{e}_{2,i} \right\|_2^2 \right),$$

$$(9)$$

where $\mathbf{e}_{1,i}$ is the code embedding obtained by querying the first-layer codebook with the latent representation $\mathbf{h}_i$, and $\mathbf{e}_{2,i}$ is obtained by querying the second-layer codebook with $\mathbf{e}_{1,i}$. The operator $\text{sg}[\cdot]$ denotes the stop-gradient operation. The terms $\mathcal{L}_{\text{vq1}}$ and $\mathcal{L}_{\text{vq2}}$ represent the vector quantization losses for the first and second layers, respectively. Finally, the total loss function is given as:

$$\mathcal{L} = \mathcal{L}_{\text{NodeRec}} + \mathcal{L}_{\text{EdgeRec}} + \alpha \mathcal{L}_{\text{vq1}} + \beta \mathcal{L}_{\text{vq2}}, \quad (10)$$

where $\alpha$ and $\beta$ are the scaling factors that control the relative contributions of the VQ losses for the first and second layers, respectively.

***Complexity Analysis.*** Assume a model depth of $L$, a codebook size of $K$, feature dimensions of $d$, and the numbers of nodes and edges denoted by $|\mathcal{V}|$ and $|\mathcal{E}|$, respectively. The time complexity and space complexity for the encoder are $O(Ld^2|\mathcal{V}| + Ld|\mathcal{E}|)$ and $O(Ld^2 + Ld|\mathcal{V}| + |\mathcal{E}|)$ respectively, which are the same as those for the node decoder. The time complexity and space complexity for the edge decoder are $O(Ld^2|\mathcal{E}|)$ and $O(d|\mathcal{E}|)$. For the vector quantization component, the time complexity and space complexity are $O(Kd^2|\mathcal{V}|)$ and $O(d|\mathcal{V}| + Kd)$. Therefore, the total time complexity and space complexity are $O\left((K + L)d^2|\mathcal{V}| + Ld^2|\mathcal{E}|\right)$ and $O\left((K + L)d^2 + d|\mathcal{E}| + Ld|\mathcal{V}|\right)$ respectively. Thus, when parameters are fixed, the complexity is linear to the number of nodes and edges, enabling the model to scale effectively to larger datasets. To further facilitate scalability, we adopted the sampling strategy used in [10] for large graph datasets. This strategy divides the graph into multiple batches for training, ensuring that the model can handle larger datasets efficiently.

## 5  EXPERIMENTS

### 5.1  Experimental Settings

***Datasets.*** We evaluate HQA-GAE on eight undirected and unweighted graph datasets, including citation networks: Cora, CiteSeer, PubMed [33]; co-purchase networks: Computers, Photo [34]; co-author networks: CS, Physics [34]; and an OGB network: ogbn-arxiv [16]. These datasets present distinct structural characteristics and feature distributions, providing a robust evaluation of our model's generalization.

***Baselines.*** We compare HQA-GAE against two major categories of self-supervised graph learning approaches: (i) contrastive learning methods, including DGI [50], GRACE [50], GIC [25], GCA [51], MVGRL [13], and BGRL [37]; (ii) autoencoding-based methods, including GAE [20], VGAE [20], SeeGera [23], ARGA [27], AGVGA [27], GraphMAE [15], GraphMAE2 [14], MaskGAE [22], S2GAE [36], and Bandana [49].

***Reproducibility.*** All results are reported as the mean and standard deviation over 10 independent runs. Further details on dataset statistics, runtime environments, hyperparameter configurations, and baseline result sources are provided in Appendix A.

### 5.2  Link Prediction

Link prediction is a common downstream task in graph SSL, aiming to predict the existence or likelihood of edges between node pairs. We adopt a dot-product probing approach as Bandana [49], where the dot-product operator $p_{\text{edge}} = \sigma(\mathbf{h}\mathbf{h}^\top)$ is applied to estimate edge probabilities. This form can be integrated with any graph SSL method without requiring an additional edge prediction model.

Table 1 presents the Area Under the ROC Curve (AUC) and Average Precision (AP) scores, showing that HQA-GAE surpasses all baselines across both metrics. Notably, on the Photo and Computers datasets, HQA-GAE exhibits nearly 20% higher performance compared to S2GAE and MaskGAE. We attribute this superiority to our use of vector quantization, which effectively compresses information, enabling more precise and robust modeling of relationships. In contrast, some autoencoder-based baselines may struggle with optimizing overly complex representation patterns, making them more prone to overfitting. Additionally, when comparing HQA-GAE to contrastive-based baselines, the absence of explicit topological learning objectives may limit their effectiveness. By leveraging a reconstruction loss that incorporates both node features and edge connections, our method fully captures the graph structural information, resulting in more distinct representations.

### 5.3  Node Classification

For the node classification task, we utilize the Support Vector Machine (SVM) on the learned node representations to predict labels. Instead of relying on a fixed public split, we adopt a 5-fold cross-validation approach, as followed in S2GAE [49]. This choice is motivated by our observation that different data splits can lead to significantly varying results, introducing high randomness[2].

---

[2]In experiments with MaskGAE and Bandana, we find that results from a single public split are notably affected by software versions, CUDA environments, and random seeds, which compromises reproducibility.

**Table 1: AUC (%) and AP (%) scores for link prediction across various methods. The best results for each dataset are highlighted in bold. "-" denotes unavailable results due to out-of-memory error.**

| Type | Method | Metric | Cora | CiteSeer | PubMed | Photo | Computers | CS | Physics | Avg. Rank |
|---|---|---|---|---|---|---|---|---|---|---|
| Contrastive Learning | DGI | AUC | 82.60 ± 1.51 | 73.36 ± 3.10 | 78.24 ± 1.50 | 84.30 ± 0.58 | 85.18 ± 0.67 | 89.53 ± 0.42 | 89.38 ± 0.64 | 12.64 |
| | | AP | 85.80 ± 1.39 | 80.89 ± 2.04 | 84.46 ± 0.57 | 81.50 ± 1.06 | 82.14 ± 1.23 | 89.63 ± 0.38 | 88.72 ± 0.50 | |
| | GIC | AUC | 91.81 ± 0.59 | 94.34 ± 0.74 | 91.89 ± 0.36 | 92.07 ± 0.37 | 82.87 ± 4.23 | 91.94 ± 0.57 | 91.44 ± 0.34 | 9.50 |
| | | AP | 91.60 ± 0.54 | 94.08 ± 0.87 | 91.30 ± 0.39 | 91.06 ± 0.44 | 83.43 ± 2.81 | 90.70 ± 0.90 | 90.43 ± 0.50 | |
| | GRACE | AUC | 81.80 ± 0.45 | 84.78 ± 0.38 | 93.11 ± 0.37 | 88.64 ± 1.17 | 89.97 ± 0.25 | 87.67 ± 0.10 | - | 11.00 |
| | | AP | 82.02 ± 0.50 | 82.85 ± 0.36 | 92.88 ± 0.30 | 83.85 ± 4.15 | 92.15 ± 0.43 | 94.87 ± 0.02 | - | |
| | GCA | AUC | 81.91 ± 0.76 | 84.72 ± 0.28 | 94.33 ± 0.67 | 89.61 ± 1.46 | 90.67 ± 0.30 | 88.05 ± 0.00 | - | 10.50 |
| | | AP | 80.51 ± 0.71 | 81.57 ± 0.22 | 93.13 ± 0.62 | 86.53 ± 3.00 | 90.50 ± 0.63 | 94.94 ± 0.37 | - | |
| | MVGRL | AUC | 91.10 ± 1.24 | 92.41 ± 1.66 | 93.40 ± 1.64 | 77.13 ± 3.28 | 87.25 ± 1.32 | 92.56 ± 0.61 | 91.77 ± 0.22 | 9.93 |
| | | AP | 91.51 ± 1.31 | 93.59 ± 1.48 | 93.22 ± 1.55 | 69.83 ± 3.42 | 84.41 ± 1.75 | 91.43 ± 0.88 | 90.64 ± 0.30 | |
| | BGRL | AUC | 93.79 ± 0.79 | 91.36 ± 1.06 | 95.93 ± 0.93 | 74.97 ± 6.86 | 91.43 ± 5.61 | 75.28 ± 1.51 | 75.14 ± 0.94 | 12.00 |
| | | AP | 89.85 ± 1.47 | 85.44 ± 1.53 | 94.04 ± 2.13 | 67.22 ± 5.86 | 87.68 ± 8.62 | 66.97 ± 1.37 | 66.83 ± 0.85 | |
| Autoencoding | GAE | AUC | 94.66 ± 0.26 | 95.19 ± 0.45 | 94.58 ± 1.12 | 71.45 ± 0.95 | 70.99 ± 1.03 | 93.78 ± 0.36 | 88.88 ± 1.11 | 10.43 |
| | | AP | 94.22 ± 0.39 | 95.70 ± 0.31 | 94.26 ± 1.65 | 65.99 ± 0.96 | 67.88 ± 0.82 | 89.87 ± 0.59 | 82.45 ± 1.59 | |
| | ARGA | AUC | 94.76 ± 0.18 | 95.68 ± 0.35 | 94.12 ± 0.08 | 85.42 ± 0.79 | 67.09 ± 3.93 | 95.49 ± 0.17 | 90.70 ± 1.08 | 8.14 |
| | | AP | 94.93 ± 0.20 | 96.34 ± 0.25 | 94.19 ± 0.08 | 80.58 ± 1.40 | 62.53 ± 3.17 | 92.56 ± 0.33 | 89.37 ± 1.16 | |
| | VGAE | AUC | 91.24 ± 0.48 | 94.55 ± 0.48 | 95.46 ± 0.04 | 95.61 ± 0.05 | 92.69 ± 0.03 | 87.34 ± 0.43 | 89.27 ± 0.83 | 8.14 |
| | | AP | 92.27 ± 0.43 | 95.34 ± 0.37 | 94.29 ± 0.07 | 94.63 ± 0.06 | 88.27 ± 0.08 | 80.24 ± 0.55 | 82.79 ± 1.14 | |
| | ARVGA | AUC | 91.35 ± 0.87 | 94.47 ± 0.33 | 96.17 ± 0.21 | 95.44 ± 0.14 | 92.38 ± 0.15 | 87.39 ± 0.37 | 88.96 ± 0.96 | 8.29 |
| | | AP | 91.98 ± 0.85 | 95.21 ± 0.33 | 94.81 ± 0.41 | 94.49 ± 0.12 | 88.49 ± 0.33 | 80.31 ± 0.49 | 82.38 ± 1.31 | |
| | SeeGera | AUC | 95.49 ± 0.70 | 94.61 ± 1.05 | 95.19 ± 3.94 | 95.25 ± 1.19 | 96.53 ± 0.16 | 95.73 ± 0.70 | - | 4.50 |
| | | AP | 95.90 ± 0.64 | 96.40 ± 0.89 | 94.60 ± 4.17 | 94.04 ± 1.18 | 96.33 ± 0.16 | 93.17 ± 0.53 | - | |
| | GraphMAE | AUC | 93.02 ± 0.53 | 95.21 ± 0.47 | 87.54 ± 1.06 | 75.08 ± 1.24 | 71.27 ± 0.89 | 92.45 ± 4.18 | 85.03 ± 7.16 | 11.86 |
| | | AP | 91.40 ± 0.59 | 94.42 ± 0.67 | 86.93 ± 1.01 | 70.04 ± 1.12 | 66.84 ± 1.11 | 91.67 ± 4.17 | 82.46 ± 9.33 | |
| | GraphMAE2 | AUC | 93.26 ± 1.00 | 95.26 ± 0.14 | 90.85 ± 0.91 | 73.03 ± 2.24 | 72.20 ± 2.09 | 94.57 ± 0.32 | 94.56 ± 0.81 | 9.71 |
| | | AP | 91.65 ± 0.98 | 94.36 ± 0.20 | 90.37 ± 0.92 | 68.77 ± 1.50 | 67.97 ± 1.52 | 92.76 ± 0.54 | 93.86 ± 1.09 | |
| | S2GAE | AUC | 89.27 ± 0.33 | 86.35 ± 0.42 | 89.53 ± 0.23 | 86.80 ± 2.85 | 84.16 ± 4.82 | 86.60 ± 1.06 | 88.92 ± 1.24 | 12.65 |
| | | AP | 89.78 ± 0.22 | 87.38 ± 0.29 | 88.68 ± 0.33 | 80.56 ± 3.74 | 78.13 ± 6.58 | 82.93 ± 1.63 | 88.20 ± 1.34 | |
| | MaskGAE | AUC | 95.66 ± 0.16 | 97.21 ± 0.17 | 97.19 ± 0.18 | 81.12 ± 0.45 | 76.23 ± 3.13 | 92.41 ± 0.44 | 91.94 ± 0.37 | 7.43 |
| | | AP | 95.65 ± 0.24 | 97.02 ± 0.32 | 96.69 ± 0.19 | 77.11 ± 0.40 | 71.71 ± 2.94 | 87.16 ± 0.69 | 86.33 ± 0.55 | |
| | Bandana | AUC | 95.71 ± 0.12 | 96.89 ± 0.21 | 97.26 ± 0.16 | 97.24 ± 0.11 | 97.33 ± 0.06 | 97.42 ± 0.08 | 97.02 ± 0.04 | 2.14 |
| | | AP | 95.25 ± 0.16 | 97.16 ± 0.17 | 96.74 ± 0.38 | 96.79 ± 0.15 | 96.91 ± 0.09 | 97.09 ± 0.15 | 96.67 ± 0.05 | |
| | HQA-GAE | AUC | **96.02 ± 0.11** | **97.41 ± 0.48** | **97.87 ± 0.08** | **97.91 ± 0.16** | **97.60 ± 0.20** | **97.76 ± 0.10** | **98.37 ± 0.06** | **1.00** |
| | | AP | **96.45 ± 0.16** | **97.65 ± 0.61** | **97.40 ± 0.11** | **97.42 ± 0.20** | **97.18 ± 0.25** | **97.64 ± 0.15** | **98.22 ± 0.08** | |

Table 2 shows that HQA-GAE achieves superior node classification performance, outperforming baselines on 6 out of 8 datasets, demonstrating its strong generalization capability across various graph structures and domain-specific features. Even on more complex or large-scale datasets like CS and ogbn-arxiv, where our method positions as the runner-up, it still maintains strong competitiveness, further highlighting its robustness and adaptability in various graph learning tasks.

## 5.4 Analysis of Core Designs

We investigate the impact of the annealing-based code selection on codebook utilization and its efficacy on downstream tasks. Additionally, we examine how the integration of the hierarchical codebook influences the quality of node representations.

*5.4.1 Effects of Annealing-Based Code Selection.* We propose this selection strategy to mitigate codebook underutilization problem caused by the argmax-based lookup function. The decayed temperature parameter shifts the code selection distribution from smooth to sharp, with the rate of this change controlled by the decay factor $\gamma$. Therefore, we vary the decay factor across {0, 0.3, 0.6, 0.9, 0.99, 0.999, 0.9999}, tracking its codebook utilization and node classification performance on three datasets. Note that a decay factor of 0 corresponds to using the conventional argmax selection without

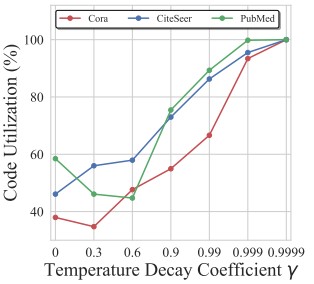

(a) Code utilization comparison

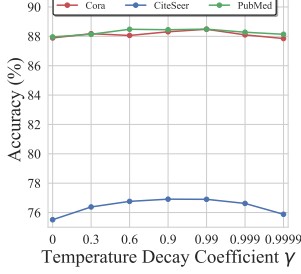

(b) Accuracy comparison

**Figure 3: The effect of annealing-based code selection.**

annealing. As shown in Figure 3(a), codebook utilization increases as the decay factor rises, eventually reaching saturation. This is expected, as a higher decay factor slows the annealing process, allowing for more exploration and leading to a more diverse use of the codebook space. For node classification performance, as depicted in Figure 3(b), it follows a similar upward trend, peaking around a decay factor of 0.9 before declining. This suggests that an appropriate codebook utilization, driven by the decay rate, alleviates selection bias and enhances downstream performance, whereas an overly

**Table 2: Accuracy (%) for node classification across various methods. The best result for each dataset is highlighted in bold. "-" denotes unavailable results due to out-of-memory error.**

| Type | Method | Cora | CiteSeer | PubMed | Computers | Photo | CS | Physics | ogbn-arxiv | Avg. Rank |
|---|---|---|---|---|---|---|---|---|---|---|
| Contrastive Learning | DGI | 85.41 ± 0.34 | 74.51 ± 0.51 | 85.95 ± 0.66 | 84.68 ± 0.39 | 91.57 ± 0.25 | 92.77 ± 0.38 | 94.55 ± 0.13 | 67.08 ± 0.43 | 10.60 |
| | GIC | 87.70 ± 0.01 | 76.39 ± 0.02 | 85.99 ± 0.13 | 82.50 ± 0.22 | 90.65 ± 0.47 | 91.33 ± 0.30 | 93.49 ± 0.42 | 64.00 ± 0.22 | 9.88 |
| | GRACE | 86.54 ± 1.50 | 73.85 ± 1.73 | 86.91 ± 0.81 | 80.75 ± 0.88 | 91.68 ± 0.93 | 92.72 ± 0.35 | 95.96 ± 0.13 | - | 8.86 |
| | GCA | 84.19 ± 1.85 | 73.81 ± 1.63 | 86.99 ± 0.68 | 88.28 ± 0.82 | 93.08 ± 1.24 | 93.66 ± 0.43 | 95.83 ± 0.16 | - | 7.57 |
| | MVGRL | 85.86 ± 0.15 | 73.18 ± 0.22 | 84.86 ± 0.31 | 88.70 ± 0.24 | 92.15 ± 0.20 | 92.87 ± 0.13 | 95.35 ± 0.08 | 68.33 ± 0.32 | 9.50 |
| | BGRL | 86.16 ± 0.20 | 73.96 ± 0.14 | 86.42 ± 0.18 | 90.48 ± 0.10 | 93.22 ± 0.15 | 93.35 ± 0.06 | 96.16 ± 0.09 | 71.77 ± 0.19 | 6.38 |
| Autoencoding | GAE | 81.81 ± 1.72 | 59.34 ± 4.75 | 83.30 ± 0.77 | 88.64 ± 0.80 | 92.59 ± 0.85 | 85.54 ± 2.59 | 93.93 ± 0.83 | - | 13.10 |
| | ARGA | 80.76 ± 1.52 | 66.76 ± 1.64 | 79.88 ± 0.58 | 80.19 ± 0.96 | 88.76 ± 0.70 | 91.86 ± 0.50 | 95.03 ± 0.16 | 58.13 ± 0.78 | 14.10 |
| | VGAE | 83.48 ± 1.55 | 67.56 ± 2.03 | 81.34 ± 0.97 | 90.35 ± 0.75 | 93.28 ± 0.76 | 83.96 ± 1.75 | 94.90 ± 0.58 | - | 12.10 |
| | ARVGA | 85.86 ± 0.72 | 73.10 ± 0.86 | 81.85 ± 1.01 | 83.36 ± 0.43 | 86.55 ± 0.31 | 84.68 ± 0.26 | 92.89 ± 0.11 | 50.06 ± 1.21 | 13.80 |
| | SeeGera | 87.70 ± 1.13 | 75.82 ± 1.67 | 85.36 ± 0.69 | 87.95 ± 1.39 | 91.88 ± 0.53 | **94.69 ± 0.21** | 88.25 ± 2.58 | - | 8.14 |
| | GraphMAE | 85.45 ± 0.40 | 72.48 ± 0.77 | 85.74 ± 0.14 | 88.04 ± 0.61 | 92.73 ± 0.17 | 93.47 ± 0.04 | 96.13 ± 0.03 | 71.86 ± 0.00 | 8.63 |
| | GraphMAE2 | 86.25 ± 0.78 | 74.68 ± 1.88 | 86.94 ± 1.49 | 74.48 ± 0.66 | 85.88 ± 0.51 | 90.59 ± 0.60 | 86.30 ± 0.15 | 72.46 ± 0.28 | 10.60 |
| | S2GAE | 86.15 ± 0.25 | 74.60 ± 0.06 | 86.91 ± 0.28 | 90.94 ± 0.08 | 93.61 ± 0.10 | 91.70 ± 0.08 | 95.82 ± 0.03 | 72.02 ± 0.05 | 6.25 |
| | MaskGAE | 87.31 ± 0.05 | 75.20 ± 0.07 | 86.56 ± 0.26 | 90.52 ± 0.04 | 93.33 ± 0.14 | 92.31 ± 0.05 | 95.79 ± 0.02 | 70.99 ± 0.12 | 6.13 |
| | Bandana | 88.59 ± 1.35 | 74.85 ± 1.51 | 88.16 ± 0.57 | 91.52 ± 0.83 | 93.64 ± 0.83 | 93.57 ± 0.12 | 96.48 ± 0.12 | **73.87 ± 0.18** | 2.50 |
| | HQA-GAE | **88.78 ± 1.03** | **76.76 ± 1.24** | **88.49 ± 0.53** | **91.79 ± 0.88** | **93.84 ± 0.75** | 94.25 ± 0.28 | **96.81 ± 0.23** | 72.93 ± 0.65 | **1.25** |

**Table 3: Comparison of hierarchical codebook and single-layer codebook on node clustering.**

| Dataset | 2-layer Codebook | | | 1-layer Codebook | | |
|---|---|---|---|---|---|---|
| | NMI | ARI | SC | NMI | ARI | SC |
| Cora | **0.544** | **0.501** | **0.189** | 0.529 | 0.442 | 0.184 |
| CiteSeer | **0.423** | **0.423** | **0.130** | 0.418 | 0.415 | 0.079 |
| Pubmed | **0.271** | 0.254 | **0.263** | 0.268 | **0.258** | 0.260 |
| Computers | **0.435** | **0.257** | **0.120** | 0.422 | 0.256 | 0.113 |
| Photo | **0.620** | **0.529** | **0.338** | 0.603 | 0.505 | 0.305 |
| CS | **0.734** | **0.612** | **0.257** | 0.722 | 0.571 | 0.253 |
| Physics | **0.642** | **0.607** | 0.177 | 0.642 | 0.587 | **0.180** |

high decay factor introduces excessive randomness, disrupting the convergence of the code embedding space.

*5.4.2 Role of the Hierarchical Codebook.* Our hierarchical codebook design aims to promote the correlation between codes, reflecting the interconnected nature of nodes in graph data. To evaluate its effectiveness, we compare the clustering capability of the hierarchical (two-layer) codebook against a single-layer codebook on seven datasets. Specifically, node representations learned from both codebook designs are subjected to k-means clustering [2, 11], and clustering quality is measured by Normalized Mutual Information (NMI) [32], Adjusted Rand Index (ARI) [29], and Silhouette Coefficient (SC) [31]. Table 3 shows that our hierarchical codebook generally outperforms the single-layer version, demonstrating superior clustering performance. This indicates that the hierarchical structure brings similar nodes closer together while increasing the separation between dissimilar ones, providing a more informative representation foundation for downstream tasks.

We further visualize the embeddings from both layers of the hierarchical codebook, along with the node representations from the encoder using t-SNE [39] in Figure 4. We observe that the second-level codes serve as cluster centers for the first-level ones, and the node representations from different classes are more clearly separated after training. This highlights that the hierarchical codebook's ability to learn expressive representations.

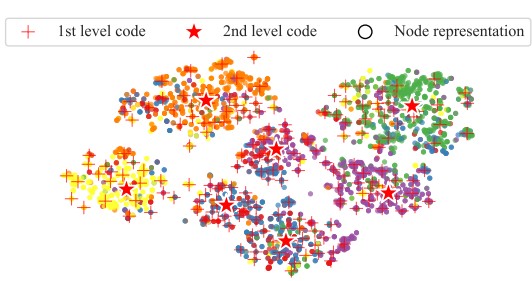

**Figure 4: Visualization of node representations and codebook embeddings on CiteSeer by t-SNE.**

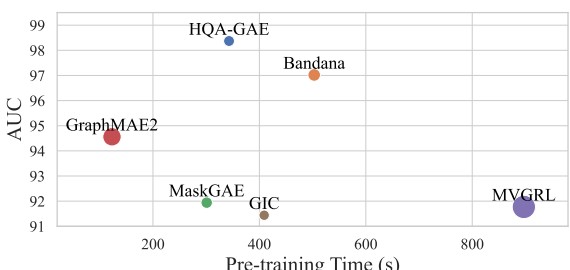

**Figure 5: Comparison of performance, model size, and pre-training time for link prediction on the Physics dataset. Each point's size represents the model size.**

## 5.5 Efficiency Analysis

To evaluate the balance between performance, model size (number of parameters), and pre-training time, we compare the top six performing models for link prediction on the Physics dataset. As illustrated in Figure 5, HQA-GAE achieves the highest performance while maintaining a relatively small size compared to competitors. In terms of pre-training time, although GraphMAE2 and MaskGAE

train faster, GraphMAE2 has more than twice the number of parameters as our model, while MaskGAE lags behind ours in performance by nearly 10%. These results demonstrate that HQA-GAE strikes a balance between performance, model size, and runtime, offering an efficient solution without compromising effectiveness.

## 5.6 Sensitivity Analysis

*5.6.1 Codebook size.* The size of the codebooks in both layers determines the exploration space for code embeddings, thus affecting the quality of node representations.

For the first-layer codebook, we vary its size from $2^1$ to $2^{12}$ across three datasets for node classification. As shown in Figure 6, performance generally improves with increasing codebook size until it stabilizes. Datasets with more node types, like Cora and CiteSeer, require a larger codebook size ($2^8$) for optimal performance, whereas PubMed only requires $2^4$. This indicates that larger codebook sizes provide sufficient embedding space to accommodate nodes with greater label diversity, whereas datasets with fewer node types necessitate less space.

For the second-layer codebook, we fix the first-layer codebook size at $2^{10}$ and vary the second-layer size from $2^1$ to $2^8$ for clustering tasks. K-means is applied to the learned node representations, with performance measured by NMI, ARI, and SC. As shown in Figure 7, performance drops when the codebook size is smaller than the number of classes in the dataset, as nodes with different labels are forced to share the same code embedding, disrupting the independence of representations between classes. As the size becomes excessively large, although unnecessary, it does not significantly impair performance.

*5.6.2 Scaling factors $\alpha$ and $\beta$.* $\alpha$ and $\beta$ control the contributions of the VQ losses for the first and second layers, respectively. We conduct node classification experiments by varying $\alpha$ in {0, 0.5, 1.0, 1.5, 2.0} and $\beta$ in {0, 0.001, 0.01, 0.1, 1.0}. As shown in Figure 8, the optimal $\alpha$ value is 1, which is used across all datasets, indicating that the reconstruction loss and the first VQ loss hold similar importance. For $\beta$, node classification performance remains nearly unchanged across different values. Nevertheless, in clustering tasks, as shown in Figure 9, relatively small $\beta$ values (0.001, 0.01, 0.1) significantly improve clustering performance, since the second-layer codebook mitigates codebook space sparsity and enhances the quality of node representations.

Due to space limitations, we relocate the discussion on the impact of encoder design to Appendix B.

## 6 CONCLUSION

In this work, we investigate the potential of self-supervised learning with VQ-VAE applied to graph data and observe the unique advantages of vector quantization in effectively enhancing the model's ability to capture graph topology in representation learning. To better adapt VQ-VAE to graph data, we propose HQA-GAE, which employs an annealed-based code selection design to alleviate the issue of codebook space underutilization, thereby improving its generalization to diverse data distributions. Moreover, we introduce a hierarchical codebook mechanism to address the problem of codebook space sparsity, ensuring that similar embeddings are closer

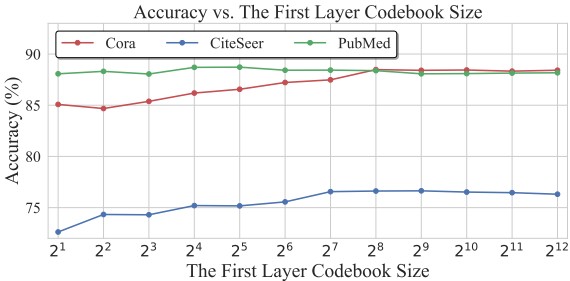

**Figure 6: The impact of varying the size of the first layer codebook**

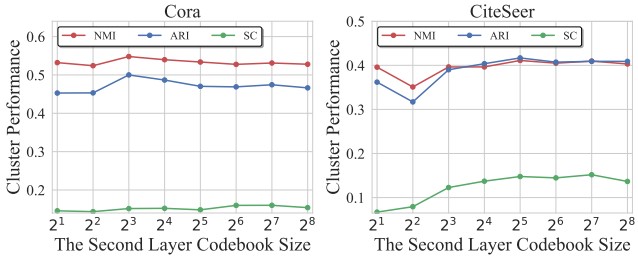

**Figure 7: The effect of the size of the second layer codebook**

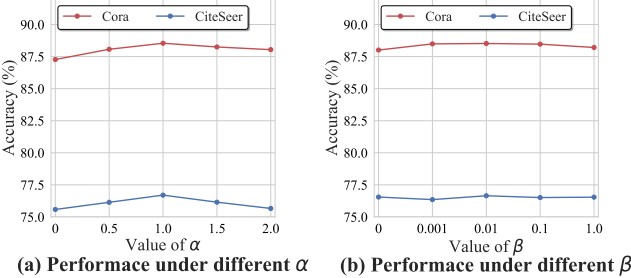

(a) Performace under different $\alpha$     (b) Performance under different $\beta$

**Figure 8: The effect of $\alpha$ and $\beta$ on node classification.**

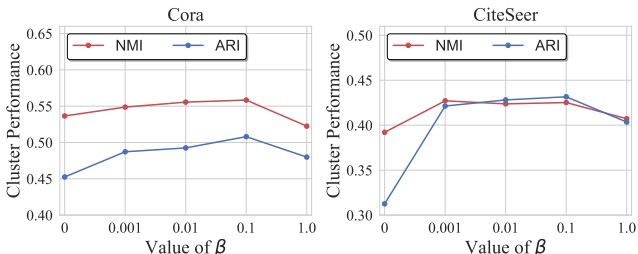

**Figure 9: The effect of $\beta$ on clustering task.**

in the representation space and exhibit improved clustering properties. Our extensive experiments and analyses demonstrate that HQA-GAE significantly outperforms all competitors on the link prediction and shows comparable performance in node classification, emphasizing the effectiveness and robutness of HQA-GAE.

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

## A    MORE CONFIGURATIONS

In this section, we provide a detailed description of our experimental setup, including the dataset statistics, hardware and software environments, and hyperparameter configurations used in the experiments.

### A.1    Data Statistics

**Table 4: Dataset statistics**

| Dataset | #nodes | #edges | #features | #classes |
|---------|--------|--------|-----------|----------|
| Cara | 2,708 | 10,556 | 1,433 | 7 |
| CitrSeer | 3,327 | 9,104 | 3,703 | 6 |
| PubMed | 19,717 | 88,648 | 500 | 3 |
| Photo | 7,487 | 119,043 | 745 | 8 |
| Cumputers | 13,381 | 245,778 | 767 | 10 |
| CS | 18,333 | 81,894 | 6,805 | 15 |
| Physics | 34,493 | 247,962 | 8,415 | 5 |
| ogbn-arxiv | 169,343 | 2,315,598 | 128 | 40 |

We use a total of 8 undirected and unweighted graph datasets, including citation networks: Cora, CiteSeer, PubMed [33]; co-purchase networks: Computers, Photo [34]; co-author networks: CS, Physics [34]; and an OGB network: ogbn-arxiv [16]. Please refer to Table 4 for detailed statistics of these datasets.

### A.2    Hardware and Software Environments

HQA-GAE is built upon PyTorch [28] 2.3.0 and PyTorch Geometric (PyG) [8] 2.5.3. The latter provides all 7 datasets used throughout our experiments except ogbn-arxiv, which is from the OGB 1.3.6 package [16]. All experiments are conducted on an 80GB NVIDIA A800 GPU with CUDA 12.1.

### A.3    Baselines

In the case of link prediction, the results for GRACE [50], GCA [51], GraphMAE [15], GraphMAE2 [14], GAE [20], VGAE [20], ARGA [27], AGVGA [27], MaskGAE [22], S2GAE [36], and Bandana are taken from Bandana [49]. Specifically, MaskGAE includes results for two variants, MaskGAE-edge and MaskGAE-path. We report the best-performing variant for comparison. For methods without published results, such as DGI [50] and GIC [25], we conduct experiments using the same evaluation setup.

Regarding node classification, the results for DGI [50], GIC [25], MVGRL [13], BGRL [37], ARVGA [27], GraphMAE [15], MaskGAE [22], and S2GAE [49] are obtained from the S2GAE paper. For methods not covered in that work, such as VGAE [20] and SeeGera [23], we conduct our own experiments using the same settings to obtain their results.

### A.4    Hyperparameter Settings

We use a GCN model with the dropout rate of 0.2 across all datasets and experiments in practice. Note that the encoder in HQA-GAE can be implemented by any GNN, we conduct an experiment to analyze the effect of different encoder models, as detailed in Appendix B. For optimization, we use the Adam optimizer with a learning rate of

**Table 5: Impact of encoder design on representations**

| Readout | Model | Cora | CiteSeer | PubMed |
|---------|-------|------|----------|--------|
| Concat | GCN | **88.62 ± 1.51** | 76.84 ± 1.65 | **88.26 ± 0.68** |
| | GAT | 88.34 ± 1.07 | 76.42 ± 1.52 | 87.07 ± 0.66 |
| | SAGE | 88.34 ± 1.46 | **76.85 ± 1.83** | 87.36 ± 0.61 |
| | GIN | 87.31 ± 1.62 | 76.55 ± 1.61 | 87.11 ± 0.68 |
| Last | GCN | 88.29 ± 1.18 | 76.21 ± 1.57 | 87.26 ± 0.67 |
| | GAT | 88.20 ± 1.25 | 75.39 ± 1.57 | 85.93 ± 0.66 |
| | SAGE | 88.00 ± 1.08 | 75.88 ± 1.65 | 85.81 ± 0.52 |
| | GIN | 86.34 ± 1.54 | 75.02 ± 2.28 | 85.37 ± 0.75 |

1e-2, except for the ogbn-arxiv dataset, where the learning rate is set to 1e-5. The edge decoder is a 2-layer MLP with a hidden size of 32. For all datasets, the scaling factor of $\mathcal{L}_{vq1}$ is set to 1. The annealing code selection strategy is applied only to the first-layer codebook, as the second-layer codebook already achieves sufficiently high utilization. The remaining hyperparameter settings are detailed in Table 6.

## B    IMPACT OF ENCODER DESIGN

We examine how the architecture of the encode influences the quality of representations.

We employ various encoders, including GCN, GAT, GraphSAGE, and GIN, across the Cora, PubMed, and CiteSeer datasets. Each encoder's output strategy is assessed by concatenating representations from all intermediate layers versus utilizing only the final layer's representation. The results are summarized in Table 5. We observe that the concatenation strategy consistently outperformed the final-layer-only strategy, likely due to the richer semantic information encapsulated in the concatenated representation, which alleviates the oversmoothing problem associated with GNNs. Furthermore, the choice of GNN as the encoder significantly influences the results, with GCN generally demonstrating superior representation capabilities.

Received 20 February 2007; revised 12 March 2009; accepted 5 June 2009

**Table 6: Detailed hyperparameters of HQA-GAE.**

| Dataset | Cora | CiteSeer | PubMed | Photo | Computers | CS | Physics | ogbn-arxiv |
|---|---|---|---|---|---|---|---|---|
| Hidden size | 512 | 256 | 256 | 256 | 256 | 64 | 256 | 1024 |
| Embedding size | 256 | 128 | 256 | 128 | 256 | 64 | 128 | 512 |
| Number of layers | 2 | 2 | 2 | 2 | 2 | 2 | 2 | 3 |
| VQ-2 loss factor $\beta$ | 0.1 | 0.1 | 0.01 | 0.01 | 0.01 | 0.01 | 0.01 | 0.01 |
| Initial temperature $T_0$ | 1 | 1 | 0.1 | 0.1 | 0.1 | 0.1 | 0.1 | 1 |
| Decay factor $\gamma$ | 0.9 | 0.9 | 0.9 | 0.9 | 0.3 | 0.3 | 0.3 | 0.99 |
| The 1st layer codebok size | 256 | 256 | 256 | 128 | 128 | 512 | 128 | 2048 |
| The 2nd layer codebok size | 16 | 32 | 8 | 16 | 16 | 64 | 32 | 128 |

