# OpenReview forum: "Hierarchical Vector Quantized Graph Autoencoder with Annealing-Based Code Selection"
_ACM.org/TheWebConf/2025/Conference — WWW 2025 Poster_

### Official Review · Reviewer_2Qbi · 2024-11-26

**Novelty:** 3
**Technical Quality:** 4

**Review:**

This paper proposes hierarchical vector quantized graph autoencoder (HVQ-GAE) for graph self-supervised learning. It claims that previous methods may generate inappropriate perturbations during view generation and HVQ can help model learn graph topology. HVQ-GAE contains a two-layer codebook to learn the hierarchical structures of graphs. Experimental results demonstrate the effectiveness of the proposed method.

Pros:
1. This paper is the first to apply VQ technology to graph self-supervised learning.

2. HVQ-GAE outperforms all baselines in link prediction task by a large margin and exhibits conpetitive results on node classficiation task.


Cons:
1. The motivation of this paper is not clear. Generally, VQ aims to learn discrete representations for data generation, such as images. However, it is unclear to me why we need VQ in graph self-supervised learning? This paper claims that inappropriate perturbations may corrupt inherent information. But how can we ensure that VQ can avoid inappropriate perturbations?

2. Existing graph autoencodes, such as GraphMAE [1] and GraphMAE2 [2], only uses node-level loss function in the reconstruction, while HVQ-GAE involves both node-level and edge-level reconstructions. This may interpret why HVQ-GAE performs well in the link prediction task. However, I am concerned if other graph autoencodes can also benefit from the edge-level reconstruction? An additional ablation is necessary to ensure the fairness of the comparison.

3. While this paper is the first to apply VQ to graph self-supervised learning, this technology has been fully explored in previous methods, such as Mole-BERT [3] and VQGraph [4], so the novelty of this paper is relatively not enough.

4. The two-layer codebook significantly increases the parameters and complexity of HVQ-GAE.

[1] GraphMAE: Self-Supervised Masked Graph Autoencoders. KDD 2022.

[2] GraphMAE2: A Decoding-Enhanced Masked Self-Supervised Graph Learner. WWW 2023.

[3] Mole-BERT: Rethinking Pre-training Graph Neural Networks for Molecules. ICLR 2023.

[4] VQGraph: Rethinking Graph Representation Space for Bridging GNNs and MLPs. ICLR 2024.

**Questions:**

Is the proposed method suitable for graph-level tasks, such as molecular property prediction?

**Reviewer Confidence:**

3: The reviewer is confident but not certain that the evaluation is correct

**Scope:**

3: The work is somewhat relevant to the Web and to the track, and is of narrow interest to a sub-community

---

### Official Review · Reviewer_aCrT · 2024-11-30

**Novelty:** 6
**Technical Quality:** 6

**Review:**

This paper explores the application of vector quantization in graph autoencoding and introduces effective improvements tailored to the unique characteristics of graphs. Notably, it is the first to propose that vector quantization, when used for graph autoencoding, has an excellent capability to capture topological structures; this claim is validated through experiments. Furthermore, the study addresses two main issues with VQ-GAE: codebook underutilization and space sparsity. To tackle these challenges, an annealing-based encoding strategy is employed to enhance model expressiveness by preventing a winner-takes-all scenario during code selection. Additionally, a two-layer encoding approach is implemented to improve representation cohesion and distinction.

Pros:
1. Motivation: The motivation is well-founded, addressing disruptions in graph representations caused by randomness in contrastive learning and VAEs. By introducing perturbation-free VQ-GAE for graph representation learning, the paper empirically demonstrates vector quantization's superior ability to capture topological structures in graph autoencoding, highlighting its unique advantages and feasibility as a novel approach.
2. Method: The paper identifies shortcomings in original VQ-GAE related to graph representation learning and addresses them using an annealing-based strategy that mitigates codebook underutilization. Furthermore, it employs a two-level encoding design to enhance node representation cohesion and address space sparsity issues. These strategies are straightforward yet well-justified.
3. Experimentation: Extensive tests on tasks such as node classification and link prediction against 16 strong baseline models show highly competitive results, particularly excelling in link prediction tasks over other models. Efficiency analysis reveals a good balance between performance and efficiency. Comprehensive ablation studies further confirm the method's rationality and effectiveness from multiple perspectives.
4. Writing Quality: The writing is clear, logically structured, and uses effective language throughout.

Cons:
1. The paper lacks an exploration of employing more levels in the codebook, which could potentially offer greater flexibility and representation power.
2. The paper's loss function is derived from VQ-VAE; however, unlike the original VQ-VAE, it lacks a weight coefficient for the second term of the VQ Loss, which may result in an inability to effectively balance the contributions of the two loss components.

**Questions:**

see Review

**Reviewer Confidence:**

4: The reviewer is certain that the evaluation is correct and very familiar with the relevant literature

**Scope:**

4: The work is relevant to the Web and to the track, and is of broad interest to the community

---

### Official Review · Reviewer_z8bS · 2024-12-01

**Novelty:** 4
**Technical Quality:** 4

**Review:**

Strengths:

1. The presentation of experimental results is comprehensive and detailed.
2. Multiple downstream tasks are used to demonstrate the effectiveness of the method.
Weaknesses:

1. The scientific problem is unclear.
2. The challenges and innovations of each method module are not clearly summarized.
3. The analysis of the research gap is logically weak.
Comments:

1. The term "inappropriate perturbations" is unclear. What exactly does "inappropriate" mean in this context? A clear definition is needed. Without this, it will be difficult to identify and analyze the scientific problem.
2. Please clearly state in the last paragraph of the introduction which experiments were conducted.
3. Please include a limitations analysis, covering error analysis and method performance.
4. In the approach section, you divide the method into three sub-modules: encoding, learning, and decoding. However, the challenges of each sub-module are not clearly summarized.

**Questions:**

1. Why do you use the term "inappropriate perturbations," which is somewhat vague? How do you define this term?
2. For Figure 1, why do you display the challenges and innovations for each module?
3. Line 92 states that VAE offers significant data compression capabilities, but can perturbation-based methods not provide this? Do you have evidence to support this claim?

**Reviewer Confidence:**

2: The reviewer is willing to defend the evaluation, but it is likely that the reviewer did not understand parts of the paper

**Scope:**

3: The work is somewhat relevant to the Web and to the track, and is of narrow interest to a sub-community

---

### Official Review · Reviewer_15fS · 2024-12-02

**Novelty:** 6
**Technical Quality:** 5

**Review:**

The paper studies the problem of graph autoencoder and vector quantization. The key novelty is an annealing-based encoding method and a two-layer codebook design. Experiments show the proposed method outperforms 16 baselines on self-supervised link prediction and node classification tasks.
Pros:
(1) Detailed hyperaprameters are reported to facilitate reproduction of experimental results.
(2) Ample experimental results with error bar and sensitivity analysis are conducted which demonstrate almost consistent improvements over all the baseline methods on 8 graph datasets. (3)
Cons:
(1)  Experiments on larger-scale graphs should be conducted to test the scalability of the proposed method.
(2) Overhead and running time analysis are missing. Similar to (1), it is unclear how well the method would perform on larger-scale datasets without such study.

**Questions:**

(1) Can the authors compare the embedding visualizations made by various methods? Since quite a few baseline methods are compared against numerically, such visualization should also be made consistently as in Figure 4. Such study would provide further evidence on why the proposed method generates more informative graph embeddings compared with baselines for self-supervised graph representation learning.

**Reviewer Confidence:**

2: The reviewer is willing to defend the evaluation, but it is likely that the reviewer did not understand parts of the paper

**Scope:**

4: The work is relevant to the Web and to the track, and is of broad interest to the community

---

### Official Review · Reviewer_a3PQ · 2024-12-02

**Novelty:** 5
**Technical Quality:** 6

**Review:**

This paper proposes two new techniques for training VQ-VAE's on graph data, using a discrete codebook instead of continuous representations for variational graph autoencoders. The first proposal is to select codes probabilistically rather than using an argmax function, with a temperature scaling parameter that sharpens the sampling distribution as training goes on. The second is to use a hierarchial codebook.

Pros:
1) Conceptually simple improvements
2) Designs yield experimental improvements and are explored experimentally in good detail

Cons:
1) Techniques not particularly justified from a graph perspective
2) Introduce some additional complexity in terms of selecting annealing rate, sizes of codebook hierarchy

The paper is well written and the techniques easy to understand and implement (an implementation is also provided). At the same time, they seem to yield good effectiveness compared to a variety of baselines on classic graph learning tasks for which graph autoencoders are used. With that said, the proposed techniques are fairly simple and purely empirically justified. Given that the motivation in this work is strictly for graph data, it would help to discuss what properties of graphs they are designed to represent. The hierarchical codebook is claimed to better represent the hierarchical nature of graph structure, but this is not demonstrated. More so, the motivation for both design choices is better utilization of the codebook, which could apply to VQ-VAEs for any data.

The proposed changes do not seem to lead to an undue increase in computational overhead. However, it does seem in practice that the selection of codebook sizes must be explored carefully for multiple levels of codebook. Moreover, the annealing rate for the codebook sampling distribution is an important parameter, and can hurt if set too high or low. These could present challenges for a practitioner in training.

**Questions:**

Could the proposed methods be used for graph generation, as VQ-VAEs are often used for generative models? Alternatively, would these techniques help VQ-VAE for other data?

**Reviewer Confidence:**

3: The reviewer is confident but not certain that the evaluation is correct

**Scope:**

4: The work is relevant to the Web and to the track, and is of broad interest to the community